# The Evolution, Genomic Epidemiology, and Transmission Dynamics of Tembusu Virus

**DOI:** 10.3390/v14061236

**Published:** 2022-06-07

**Authors:** Yongqiu Cui, Yang Pan, Jinshuo Guo, Dedong Wang, Xinxin Tong, Yongxia Wang, Jingyi Li, Jie Zhao, Ying Ji, Zhi Wu, Penghui Zeng, Jianwei Zhou, Xufei Feng, Lei Hou, Jue Liu

**Affiliations:** 1College of Veterinary Medicine, Yangzhou University, Yangzhou 225009, China; cuiyongqiu97@163.com (Y.C.); 15930270598@163.com (J.G.); wddyzu@163.com (D.W.); jingyilee00@163.com (J.L.); ylsheidong@163.com (J.Z.); jiy0725@163.com (Y.J.); wuzhi19848049073@163.com (Z.W.); zengpenghui1120@163.com (P.Z.); jwzhou@yzu.edu.cn (J.Z.); xffeng@yzu.edu.cn (X.F.); 2Jiangsu Co-Innovation Center for Prevention and Control of Important Animal Infectious Diseases and Zoonoses, Yangzhou University, Yangzhou 225009, China; 3College of Animal Science and Technology, Anhui Agricultural University, Hefei 230036, China; panyang2939@163.com (Y.P.); tongxinxin426@163.com (X.T.); mercy171208@163.com (Y.W.)

**Keywords:** Tembusu virus, evolutionary origin, phylodynamic, epidemiology, phylogeography

## Abstract

Tembusu virus (TMUV) can induce severe egg drop syndrome in ducks, causing significant economic losses. In this study, the possible origin, genomic epidemiology, and transmission dynamics of TMUV were determined. The time to the most recent common ancestor of TMUV was found to be 1924, earlier than that previously reported. The effective population size of TMUV increased rapidly from 2010 to 2013 and was associated with the diversification of different TMUV clusters. TMUV was classified into three clusters (clusters 1, 2, and 3) based on the envelope (E) protein. Subcluster 2.2, within cluster 2, is the most prevalent, and the occurrence of these mutations is accompanied by changes in the virulence and infectivity of the virus. Two positive selections on codons located in the *NS3* and *NS5* genes (591 of NS3 and 883 of NS5) were identified, which might have caused changes in the ability of the virus to replicate. Based on phylogeographic analysis, Malaysia was the most likely country of origin for TMUV, while Shandong Province was the earliest province of origin in China. This study has important implications for understanding TMUV and provides suggestions for its prevention and control.

## 1. Introduction

Tembusu virus (TMUV), a member of the genus *Flavivirus* within the family Flaviviridae [1], was first reported to associate with egg production decline and retarded growth in ducks in China in 2010 [2]. It is considered to have a very wide geographical distribution in China and Southeast Asia (Malaysia and Thailand), causing huge economic losses [3,4,5,6]. The TMUV genome contains a single positive-stranded RNA of approximately 11 kb and comprises a long open reading frame (ORF) that encodes three structural proteins, capsid (C), premembrane (prM), and envelope (E), and seven nonstructural proteins (NS1, NS2a, NS2b, NS3, NS4a, NS4b, and NS5) [7]. As the most highly variable of the three structural proteins, the E protein is related to viral adsorption and assembly and induces neutralizing antibodies [8]; thus, the *E* gene is usually used for phylogenetic analyses of genetic variation and molecular epidemiology.

Genotyping of TMUV is typically based on the use of the *E* gene for classification. To date, there have been three confirmed clusters, clusters 1, 2, and 3 [7], which may be associated with changes in virulence or vaccine immunity. Previous research has shown that the time to the most recent common ancestor (tMRCA) of TMUV was 1934 [9]. Although recombination events occur during the evolution of TMUV, more evidence suggests that strong negative selection influences the evolution of TMUV [10]. The analysis of synonymous codon usage bias also showed that the codon usage bias of all Flaviviridae viruses is mainly influenced by natural selection [11,12].

TMUV can cause severe egg drop syndrome in infected ducks, which exhibit many clinical symptoms including lack of energy, decreased appetite, and elevated body temperature. The morbidity rate of infected ducks can reach 80–90%, and the mortality rate can reach 10–30% [13,14]. Researchers have found that TMUV can infect not only ducks but other avian species, including chickens, geese, pigeons, and sparrows, suggesting that TMUV has a wide host range. Some studies have found that TMUV does not exhibit a bird species barrier and that mosquitoes can play an important role in the transmission of TMUV [9,15,16,17]. The injection of TMUV into the brains of mice has shown that the mice develop significant neurological symptoms and die [18]. A serological survey of duck factory workers in Shandong Province, China, found that approximately 70% had TMUV seropositive antibodies, and 47% of oral swabs from workers were positive for viral RNA. Similar results were observed in Thailand, suggesting that TMUV may infect humans [19,20]. Other viruses in the Flaviviridae family, such as Zika virus, West Nile virus (WNV), and dengue virus, pose a significant threat to human public health [21,22,23,24]. An in-depth analysis of the evolutionary origin and transmission pathway of TMUV is needed, similar to that for other flaviviruses. Phylogenetic analysis is a powerful tool that has been applied to trace the origins and evolution of viruses [25].

Although some researchers have used bioinformatics methods to study the origin and evolution of TMUV, the many different inference methods lack uniformity in different studies. In this study, we obtained TMUV gene sequences reported worldwide from the National Center for Biotechnology Information (NCBI) and analyzed them using maximum likelihood (ML), Markov chain Monte Carlo (MCMC), and other phylogenetic tools to study the origin, evolution rate, transmission dynamics, and geographic distribution of TMUV. Our findings provide a new global view for the prevention and control of future TMUV epidemics.

## 2. Materials and Methods

### 2.1. Sequence Datasets

All sequences used in the present study were downloaded from the NCBI GenBank database (https://www.ncbi.nlm.nih.gov/, accessed on 1 May 2022), and all available TMUV strains were obtained (accessed on 1 May 2022) (Appendix A). In addition, we downloaded 20 strains of Zika virus, 20 strains of WNV, 10 strains of mosquito flavivirus, and four strains of Quang Binh virus.

Multiple alignments were performed using the fast Fourier transform (MAFFT) algorithm to compare and sort the complete genome and viral gene sequences used in this study [26]. SplitsTree and RDP4 software were used for the recombination analysis, and the recombination events were confirmed using at least four methods, with a *p*-value cut-off of 0.05 [27].

### 2.2. Tracking the Origin of TMUV

Bayesian Markov chain Monte Carlo (MCMC) methods within Bayesian evolutionary analysis sampling trees (BEAST) (V1.10.4) are typically used to investigate tMRCA and evolutionary rates [28,29]. The best substitution models of different genes were selected using ModelFinder software in accordance with the Bayesian information criterion (BIC) and the path sampling/stepping-stone sampling (ps/ss) values [26,30]. Three independent analyses were performed with a chain length of 1 × 10^9^ generations, obtained every 1000 generations and combined using LogCombiner software. The results were estimated using Tracer software (V1.7.1) after burn-in (10%). Parameters with an effective sampling size of >200 were adopted. The final MCC tree was displayed using FigTree software (V1.4.4). A Bayesian skyline coalescent model was set to estimate the efficient population size. Three independent runs were operated with a chain length of 1 × 10^9^ generations and sampled at every 10,000 generations, and the final tree was generated by TreeAnnotator (1.10.5). The final tree was input to Tracer to generate a Bayesian skyline plot (BSP).

Bayesian stochastic variable selection (BSSVS) is typically used for spatial propagation [29,31]. According to the protocol, BSSVS allowed a Bayes factor (BF) and posterior probability (PP) test, which was used to identify the most reliable description of the spreading mode. A BF > 15 and PP > 0.5 can be considered a significant migration route between country pairs. The BF of the migration rates and PP values were calculated using SpreaD3.

To understand the relationship between different species of the Flaviviridae family, a Bayesian inference (BI) tree was constructed with hosts of different species of the *E* gene using MrBayes software with the GTR + I + G + F model with two parallel runs of 2,000,000 generations; the initial 25% of sampled data were discarded as burn-in [32]. An ML tree of the TMUV *E* gene was constructed using RAxML software based on the GTRGAMMAI model with 1000 replicates.

### 2.3. Selection Analysis of TMUV

The TMUV genome was selected using DATAMONKEY (http://www.datamonkey.org/, accessed on 1 May 2022). The methods used to estimate the positive codon sites included fixed-effects likelihood (FEL), fast unconstrained Bayesian approximation (FUBAR), single-likelihood ancestor counting (SLAC), and the mixed-effects model of evolution (MEME). Positive selection sites were confirmed by at least two methods, with *p*-values for both FEL and MEME of less than 0.05, *p* < 0.1 by SLAC, and a posterior probability > 0.9 by FUBAR.

### 2.4. Structural Simulation of the TMUV E Proteins

The E protein structures of different TMUV clusters were built by homology modeling, and suitable templates were selected for these protein sequences using Alphafold2 for homology modeling [33,34]. PyMOL was used to generate model images.

## 3. Results

### 3.1. Evidence for the Temporal Origin of Emerging TMUV Worldwide

A total of 110 complete genome sequences, representing different clusters of TMUV, were aligned to analyze the potential recombination events of TMUV. The results indicated a recombination event in KJ740746, after which it was removed from the BEAST analysis. To better determine the evolutionary relationships among the different TMUV strains, a Bayesian MCMC estimation of the tMRCA was performed. Under the best clock model, the tMRCA of the complete genome was 1924, with a 95% highest probability density (HPD) ranging from 1850 to 1983. The tMRCA for the other genes was concentrated between 1932 and 1982 (Figure 1a). The nucleotide substitution rate of the complete genome was 9.66 × 10^−4^ substitutions/site/year with 95% HPD ranging from 5.49 × 10^−4^ to 1.46 × 10^−3^ substitutions/site/year. The other gene substitutions ranged from 1.29 × 10^−3^ to 1.8 × 10^−3^ substitutions/site/year (Figure 1b). Our predicted substitution rate was faster than that previously predicted (5.9 × 10^−4^ substitutions/site/year) [10].

Bayesian skyline plot (BSP) analysis results can indicate effective population size changes in TMUV. We identified that the total TMUV showed a constant population size until approximately 15 years ago (2008), when a sharp increase in population size was observed, lasting for approximately five years (2013), after which the population remained high (Figure 2). When the TMUV effective population experienced rapid expansion, the TMUV population also experienced rapid differentiation, forming multiple clusters.

### 3.2. TMUV Genomic Epidemiology Analysis Based on Phylogenetic Analysis

To better understand the epidemiology of TMUV, we constructed an MCC tree and an ML tree based on the *E* gene to further analyze the trend and classification of TMUV. As shown in Figure 3a,b, three independent clusters of TMUV were observed in both trees and showed the same structures (clusters 1, 2, and 3). Cluster 2 can be divided into two subclusters, 2.1 and 2.2. The MCC and ML trees (Figure 3a,b) clearly show that the TMUV virus originated from cluster 3 and then formed clusters 1 and 2. Phylogenetic analysis showed that cluster 2 is the most prevalent strain in China while cluster 1 is only found in Malaysia. Furthermore, according to the phylogenetic analysis, subcluster 2.2 is the dominant strain in cluster 2; other avian TMUV strains, such as sparrow TMUV, are only found in subcluster 2.2, indicating that only subcluster 2.2 has the ability to infect other avian species. This diversity in viral genotypes may be associated with altered virulence.

To further analyze the evolutionary relationship between different types of TMUV, we aligned the sequences of E proteins of different clusters of TMUV. We identified three specific amino acid mutation sites that may be key sites participating in the evolution of TMUV from cluster 3 to clusters 1 and 2 (Figure 4). Cluster 3 has a P73S mutation existing in the branch of cluster 1. Next, the S83P mutation appears to belong exclusively to cluster 2. Cluster 2 is further divided into two subclusters, 2.1 and 2.2, due to mutation of asparagine to serine at position 277 of the E protein. Mutations in this amino acid may alter the virulence and transmission of the virus. The mutation of the 83rd amino acid of the E protein to proline may enhance the transmissibility of TMUV, which may also explain why cluster 2 is more widespread. To better classify TMUV in the future, it will be necessary to combine phylogenetic analyses and amino acid mutation sites.

### 3.3. The Process of Global Dissemination of TMUV

To eliminate uncertainty in the estimation process, the relationships among the strains were estimated based on well-supported contacts (BF > 15 and PP > 0.5) in different countries.

As shown in Figure 5a and Appendix A, Malaysia was found to be the earliest location of the original TMUV, consistent with a previous study in Malaysia in which TMUV was first identified in mosquitoes in the 1960s [35,36]. Shandong Province was the first location where TMUV strains were discovered in China and based on the phylogeographic analysis results was most likely responsible for the spread of the virus to other provinces.

To further analyze the distribution of TMUV in Asia, an MCC tree was constructed (Figure 5b). In addition to Shandong Province, we found that several provinces with a developed poultry industry also had a higher incidence of TMUV, including Fujian, Jiangsu, and Anhui provinces. The transmission route found in this study is consistent with the incidence of TMUV in China, that is, it first occurs in coastal areas and then continues to spread inland [20].

### 3.4. Relationship between Different Members of the Flaviviridae Family

To understand the relationships among different members of the Flaviviridae family, we downloaded complete genomic sequences of Zika virus, dengue virus, WNV, Quang Binh virus, and mosquito flavivirus to reconstruct BI trees (Figure 6). We found that TMUV showed a close relationship with WNV (bootstrap = 100%) and Zika virus (bootstrap = 79.8%), and they were in the same clade, indicating that TMUV is closely related to WNV and Zika virus. Meanwhile, it was found that mosquito flavivirus and Quang Binh virus also showed a close relationship (bootstrap = 100%). Based on the phylogenetic analysis results, mosquito flavivirus might be the common ancestor of TMUV, WNV, and Zika virus, indicating that mosquitoes play an important role in the spread of flaviviruses.

### 3.5. Selection Analysis of TMUV NS3 and NS5 Protein

A total of 110 coding sequences of TMUV isolates (Appendix A) were used to determine whether selection pressure led to variation. All genes of TMUV were examined, but only two positive selection sites were identified with high confidence, the *NS3* and *NS5* genes (position 591 of the NS3 protein, position 883 of the NS5 protein), as shown in Table 1. These two positive codons occurred in the NS3 and NS5 proteins, respectively, and were further confirmed using at least two of the four methods (FEL, FUBAR, SLAC, and MEME). The NS3 and NS5 proteins, which serve as major enzymatic components of the viral replication complex, promote efficient viral replication through their close binding to cellular host factors [37,38,39]. These positive selection sites may lead to the altered replication capacity of TMUV.

## 4. Discussion

TMUV was first identified in mosquitoes in Malaysia in the late 1960s [36]. It was not detected again until April 2010, in an outbreak in Chinese coastal provinces, after which it quickly spread to various poultry provinces in China, including Fujian, Guangdong, and Anhui provinces [10,40,41,42]. In view of the harmful effects of TMUV, most researchers have focused on its pathogenic mechanisms [43,44]. Herein, we provide new insights into the origin time, genome epidemiology, transmission dynamics, and selection pressure of TMUV, and we analyze the phylogeographic distribution of TMUV for the first time.

Currently, tMRCA is the most reliable method for calculating the time of origin [45]. After testing all the computational models, we selected the best optimal model (GTR + G + F + I based on the Bayesian skyline coalescent model, assuming an uncorrected relaxed clock (lognormal)) to compute the most recent common ancestor time of TMUV. The tMRCA of TMUV, based on the complete sequence, was 1924 (95% HPD: 1850–1983). The tMRCA times calculated based on the different genes were all concentrated in the range of 1932–1982 (Figure 1a). The most recent common ancestor time we calculated was approximately 40 years earlier than the earliest detection in mosquitoes and nearly 90 years earlier than the detection in duck flocks, indicating that TMUV was widespread among mosquitoes before its discovery. Based on the above results, we believe that TMUV has evolved the ability to infect avian species after a long period of adaptation, following transmission to the mosquito population. TMUV was identified in 2010 in China and in the following three years, with the absence of vaccines and effective control measures, the effective population size of TMUV experienced rapid expansion, and the virus spread to many provinces and cities in China (Figure 2). Since 2013, the population of TMUV has shrunk slightly before stabilizing (Figure 2), which may be related to the use of TMUV vaccines and the continuous implementation of strict prevention and control measures.

The evolutionary rate of viruses can often be represented by changes in the rate of nucleotide mutations. Mutations occur because of the lack of proofreading capabilities of RNA-dependent RNA polymerase of the RNA virus, producing quasispecies that allow the rapid adaptation of the virus to a new host. In this study, we calculated the evolution rate of the TMUV using Bayesian methods. The nucleotide mutation rate of the *E* gene was 1.68 × 10^−3^ substitutions/site/year (95% HPD = 9.44 × 10^−4^ − 2.59 × 10^−3^ substitutions/site/year) (Figure 1b), faster than that previously reported [11]. The full-genome sequences and other genes also have considerably fast mutation rates (Figure 1), which may accelerate the evolution of the virus, it’s adaptation to new host, and so on. There are several reasons why the evolutionary rate presented in the current study is faster than that calculated in previous studies. First, the TMUV vaccine has been in use in recent years. The use of vaccines may accelerate the nucleotide mutation of TMUV by evading the host immunity, as seen with the hepatitis B virus [46]. Second, only the evolution of TMUV in ducks was investigated in previous studies, but TMUV has been found in a variety of birds and even mammals, including humans and mice [19,47]. Therefore, the evolution rate of TMUV will be faster than that predicted previously. Adaptive evolution affects certain sites of viral genes because of strong functional constraints. In this study, the results of the selective pressure analysis showed two positive sites located in nonstructural genes (position 591 of the NS3 protein and position 883 of the NS5 protein). The nonstructural proteins of the virus are usually involved in virulence, replication, and other functions. These mutations may cause changes in the virulence of the virus.

The classification of circulating viral strains will facilitate better vaccine development and the prevention and control of viral infections. In this study, we used the *E* gene to divide TMUV into three clusters and found that cluster 2 is the most prevalent genotype in China at present and that the strains that can infect chickens, pigeons, and mice are all in cluster 2 (Figure 3). The *E* gene is associated with the adsorption and assembly of TMUV; it also mediates fusion with the host cell membrane and induces the production of neutralizing antibodies [48]. To better analyze the typing of TMUV strains, we compared the E protein sequences of different TMUV clusters. Interestingly, we identified some amino acid sites that may be related to the evolution of TMUV (Figure 4). Changes in these amino acid sites often lead to changes in virulence and may lead to cross-species transmission events; this phenomenon is very common in influenza virus [49]. In this study, we found that the amino acid mutations are all neutral amino acid mutations; thus, determining whether they will mutate into acidic or basic amino acids in the future, and the possible effects of the mutation, requires further research. In addition, based on our analyses (Figure 3) we identified the three oldest strains of TMUV from mosquitoes in Malaysia and *Anas platyrhynchos* in Shandong Province, China. Although the discovery of these three strains was later than the time when TMUV was first reported, according to the results of the phylogenetic analysis, these three strains still represent the oldest TMUV strains, indicating that in the early stage of TMUV transmission, mosquitoes and ducks are the main targets of infection before gradually spreading to other animals.

Tracing the source of the virus transmission route will help prevent and control its spread in the future [24,50]. To better understand how TMUV spreads, we reconstructed the migration route of TMUV. The phylogeographic analysis results showed that Malaysia was the country of origin with high BF and PP values (Figure 5a, Table 1), which was consistent with previous results [36]. Malaysia is a tropical country with high temperature and rainfall year-round. It has a tropical rainforest oceanic climate, with four distinct seasons. Malaysia’s unique climate is conducive to the survival and spread of viruses, and the country is rich in animal species. Malaysia has a variety of bats and mosquitoes that can transmit viruses; for example, mosquito-borne viruses, such as dengue and Zika, are present in Malaysia [51,52]. Therefore, we believe that Malaysia is the most likely country of origin for TMUV. China is currently the country with the most severe TMUV epidemics. Since the virus was discovered in China’s coastal provinces in 2010, it has spread to inland provinces over a short period of time. In the phylogeographic analysis of TMUV, we found that Shandong was the first province in China where the virus appeared and it then spread to nearby provinces, such as Jiangsu, Fujian, and Hebei Province, through various means before spreading throughout the country. According to the data provided by the Ministry of Agriculture and Rural Affairs of the People’s Republic of China (http://www.moa.gov.cn/, accessed on 1 May 2022), as of 2018 the annual output of commercial ducks in Shandong Province accounted for approximately one third of the national total, and the inventory of laying ducks accounted for approximately one tenth of the national total, occupying an important position in China’s duck industry. Several other provinces, including Jiangsu, Anhui, and Guangdong, have large poultry industries. This large-scale transportation of poultry and intensive farming has contributed to the transmission rate of TMUV, allowing it to quickly spread across the country. However, we cannot ignore the possibility that TMUV can be transmitted by birds, such as pigeons; there is already evidence that it can be transmitted in waterfowl and other birds [16,17], which will create huge challenges in its prevention and control. We also need to be aware that this mode of transmission is similar to that of influenza viruses and may pose potential public health risks [53]. In addition, there is a crucial question that needs to be addressed by researchers: how did TMUV spread from Malaysia to China? The first possibility is that mosquitoes carrying TMUV entered China in some way, perhaps through commodity trade or population movement, and that cross-species transmission to birds occurred, resulting in a TMUV epidemic. The second possibility is that the mosquito TMUV in Malaysia was transmitted to birds, and their migration to China resulted in the occurrence of TMUV infection in Chinese flocks. Of the above two possible transmission routes, we believe that the second route is more likely because based on the results of the phylogenetic analysis, the TMUV carried by birds is more homologous to the TMUV of birds. However, we cannot rule out the possibility of both routes of transmission.

Members of the Flaviviridae family pose a huge threat to global public health, but their transmission and pathogenic mechanisms are not clearly understood [21,23]. Phylogenetic analysis results indicated that TMUV has a close relationship with Zika virus and WNV, with high bootstrap support (Figure 6). Based on phylogenetic analysis, we found that they originated from mosquito flavivirus and Quang Binh virus (Figure 5), and the Quang Binh virus was shown to be a type of mosquito flavivirus [54]. In addition, we found that TMUV strains and duck TMUV strains were present in the oldest genotype (cluster 3). The TMUV strains found in other avian species (chicken, goose, sparrows, and pigeons) were present only in cluster 2 (Figure 3a). This result indicates that TMUV may first be transmitted to ducks through mosquitoes and then through a variety of transmission routes, infecting different birds and even mammals. Therefore, mosquitoes may be intermediate hosts in the transmission of TMUV and mosquito flaviviruses may be the origin of TMUV, after evolution into a strain that was transmissible to birds. Mosquitoes, as important vectors of TMUV transmission, may pose huge risks in the future, and Zika virus is an example worthy of our attention. Researchers in China have found that a change in the 106th amino acid position of the Zika virus capsid protein increases the spread of Zika virus; the amino acid substitution facilitates the maturation of structural proteins and the formation of infectious viral particles [55]. The mutation rate of the capsid protein of TMUV reaches 1.81 × 10^−3^ substitutions/site/year, which is relatively fast in virus evolution, and we identified potential positive mutation sites on the NS3 protein. The two viruses (Zika virus and WNV) most closely related to TMUV are zoonotic viruses that pose a threat to public health, and some serological investigations have found TMUV antibody-positive cases in humans. Because TMUV is still evolving, and different flaviviruses are constantly interacting in the natural environment, human infection by TMUV is probably inevitable. It is therefore vital that prevention and control measures against TMUV be improved [21,56].

In conclusion, we successfully calculated the tMRCA and base substitution rate of TMUV and confirmed that avian TMUV is formed after multiple spills of the mosquito flavivirus and that TMUV can be transmitted by birds. Malaysia is thought to be the origin of TMUV, and from there it subsequently spread to China. Although TMUV is currently defined as a virus that infects only avian species, it cannot be ruled out as a potentially zoonotic virus. Overall, we performed a comprehensive bioinformatics analysis of TMUV, which should facilitate the planning of effective control strategies for TMUV infections in the future.

## Figures and Tables

**Figure 1 viruses-14-01236-f001:**
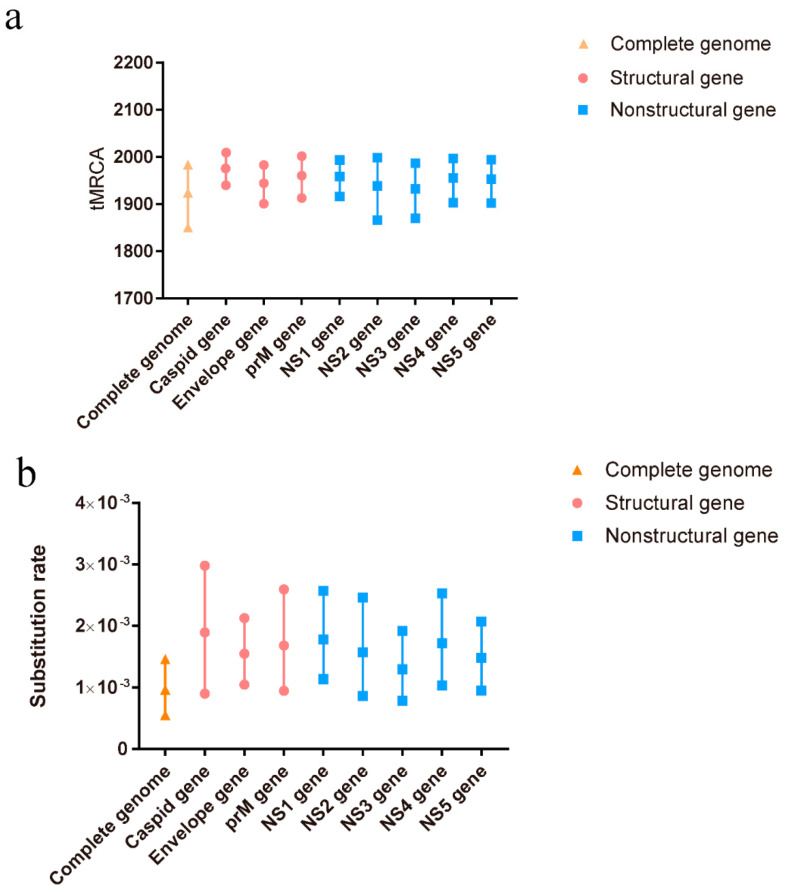
The tMRCAs and substitution rates were evaluated in BEAST (v1.10.4) for the complete genome and individual genes of TMUV. (**a**) The tMRCA of TMUV. (**b**) The substitution rate of TMUV. Yellow triangles represent the complete genome, red circles represent structural genes, and blue squares represent non-structural genes.

**Figure 2 viruses-14-01236-f002:**
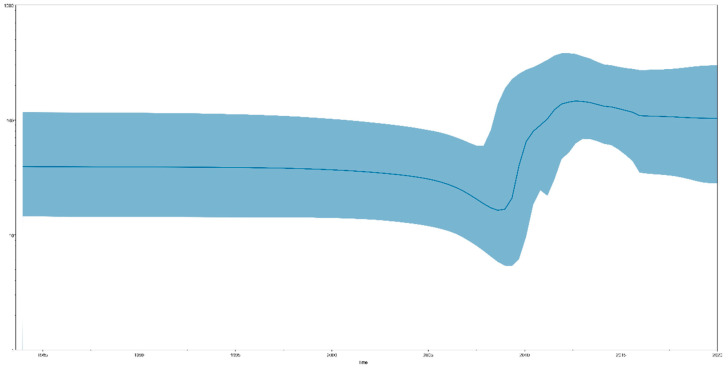
A Bayesian skyline plot exhibiting the effective population size change for the *E* gene of TMUV. A measure of group change is shown on the *y*-axis with 95% HPD (blue).

**Figure 3 viruses-14-01236-f003:**
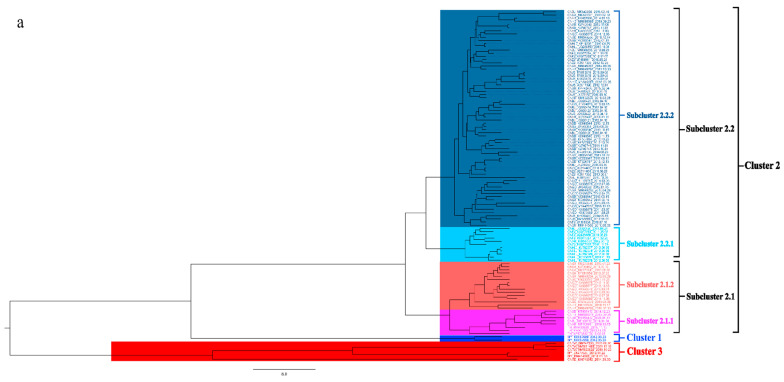
Phylogenetic analysis of the *E* gene sequences based on two different methods. (**a**) The MCC tree was scaled to time using the GTR + F + G substitution model and an uncorrected relaxed clock (lognormal) of the TMUV *E* gene. The different RGB colors represent different clusters. The RGB color numbers were 0740F9, FF33FF, FF6666, 00CCFF, 1D5792, and FF0000 for clusters 1, 2.1.1, 2.1.2, 2.2.1, 2.2.2, and 3, respectively. (**b**) The ML tree was reconstructed using RAxML based on GTR + F + I + G4 model.

**Figure 4 viruses-14-01236-f004:**
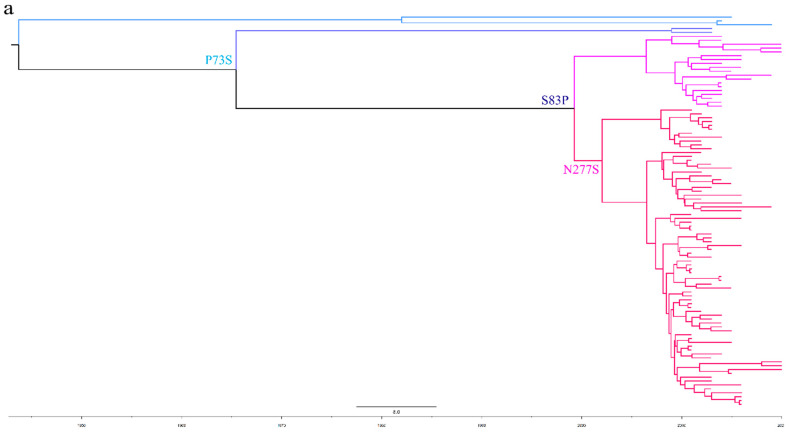
The phylogenetic analysis of the E amino acid sequence and a schematic diagram of the structure simulation of the E protein of different clusters of TMUV using AlphaFold2. (**a**) The ML tree of the E protein of TMUV and the characteristic amino acid sites are also marked with different colors. (**b**) Cluster 1 characteristic amino acid sites. (**c**) Subcluster 2.1 characteristic amino acid sites. (**d**) Subcluster 2.2 characteristic amino acid sites. (**e**) Cluster 3 characteristic amino acid sites.

**Figure 5 viruses-14-01236-f005:**
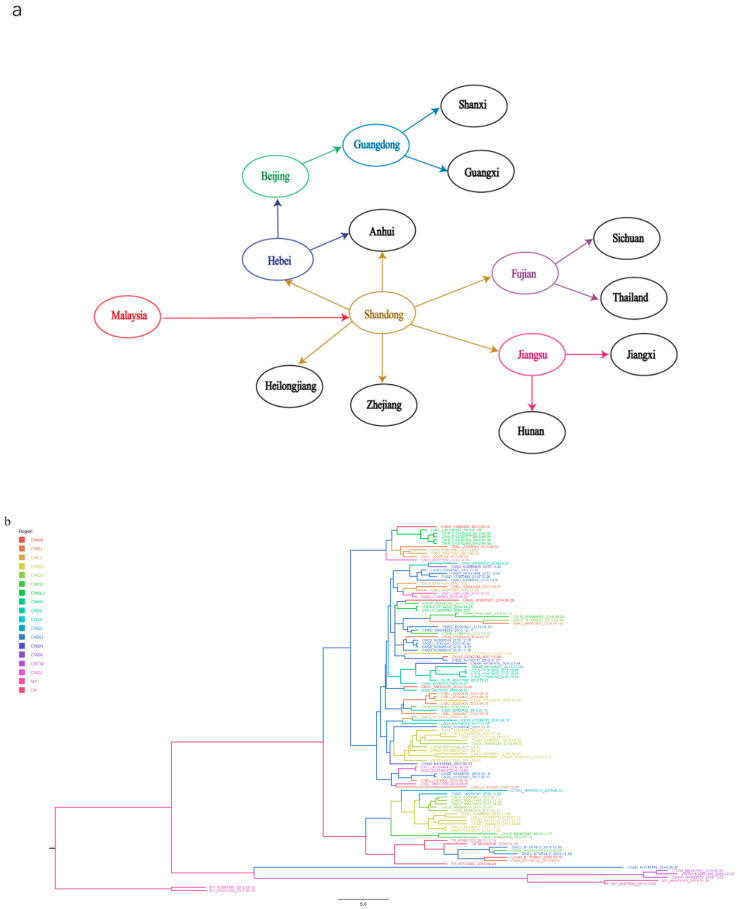
TMUV spread routines among different countries resolved using BSSVS. (**a**) The discrete phylogeographic analysis was performed with the Bayesian stochastic search variable selection (BSSVS) approach, for which we displayed the intensity of the transition rates associated with a BF > 15 and PP > 0.5. (**b**) Based on evolutionary analysis constructed from phylogeography, different regions are marked with different colors.

**Figure 6 viruses-14-01236-f006:**
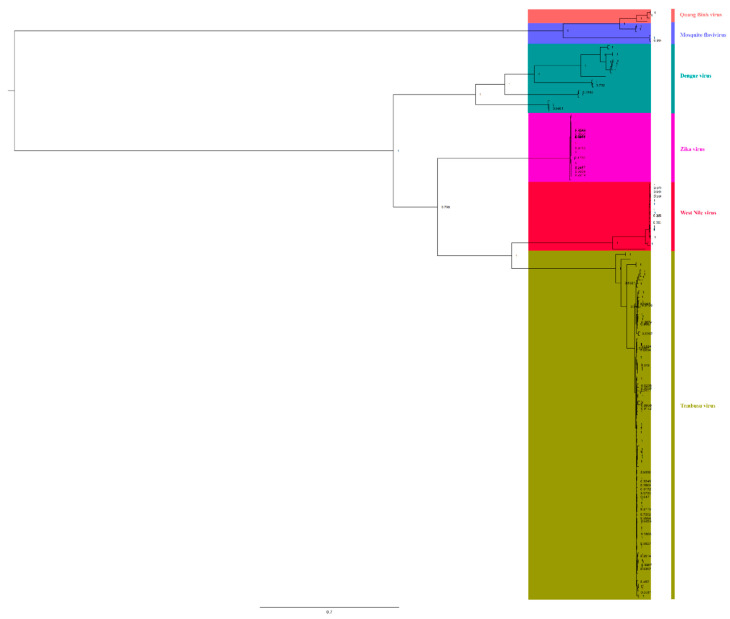
The relationship of TMUV was deduced using the *E* genes of the different flaviviruses. The BI tree was reconstructed using Mrbayes. The different flavivirus are expressed by different RGB colors, as indicated. The RGB color numbers are FF6666, 6666FF, 009999, FF00CC, FF003C, and 999900, for Quang Binh virus, mosquito flavivirus, Zika virus, Dengue virus, West Nile virus, and Tembusu virus, respectively.

**Table 1 viruses-14-01236-t001:** The selection analysis of NS3 and NS5 sequences of TMUV.

Codon	FEL		SLAC		FUBAR		MEME	
	dN-dS	*p*-Value	dN-dS	*p*-Value	dN-dS	Post. Pro	W^+^	*p*-Value
NS3-591	3.279	0.072	2.42	0.304	19.058	0.825	3.28	0.09
NS5-883	1.302	0.11	3.16	0.226	2.394	0.991	4.41	0.05

The value represents *p* < 0.1 and posterior probability > 0.9, with a significant difference.

## Data Availability

The data supporting the findings of this study are available upon request from the corresponding author.

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
