# Peer review of "The Evolution, Genomic Epidemiology, and Transmission Dynamics of Tembusu Virus"

_viruses, 2022, doi:10.3390/v14061236_

Round 1

Reviewer 1 Report

The authors used state of the art methods for their purpose correctly and indeed this kind of study could be a good participation to Tembusu virus field. But one thing is crucial, these kinds of studies must be discussed really carefully thinking that is the given result reflects the real situation or what are the chances that the given result is due to lack of more data. To be more clear, there is a gap between 1960 virus isolation in Malaysia and 2010 outbreak in China, even the first isolation of the virus from mosquitoes was 2013-2015. Considering the time gap and lack of studies in the wild, I doubt that the virus was not in an unidentified country or vector or host. However, the main issue with this manuscript is the language. The presentation of the methods, results, and figures are highly confusing and misleading. The explanation why the chosen methodology was followed is not clear every time. Especially, in the results section the language needs major changes including re-addressing the wrong interpretations. Lastly, the authors should revisit their language, explain well their results in accordance with the literature and interpret the results carefully. I recommend to the authors at least to check the following similar paper which is very well written “Genetic diversity and molecular evolution of human respiratory syncytial virus A and B”.

Reviewer 2 Report

It is a 'bioinformatics' manuscript that clarifies and expands our knowledge about Tembusu virus (TMUV), since the authors use sequences of a larger number of TMUV strains than previously. The article may be of interest to readers and should be published. The topic of the article is relevant, the article contains interesting results.

My main complaints are about the interpretation of the results, the quality of the illustrative material, and English.

Major remarks.

  1. Selection of TMUV strains whose sequences were used is unclear. Currently, there are more than 160 sequences of complete genomes or coding regions of TMUV in GenBank. The authors use only ~100 sequences. They removed from the study the most interesting TMUV sequences from Taiwan (GenBank MW922032, MW821486, MN747003, etc.). These sequences are separated from the common root with West Nile sequences even earlier than the sequences from the cluster 3, which is noted by the authors. When authors include additional available TMUV sequences, the hypothesis of the TMUV origin in Malaysia may crumble. So, the authors should add TMUV sequences from Taiwan and check their hypothesis about the place of the origin of the virus.
  2. The authors confuse “the time of the origin of a virus” and tMRCA, which is the estimating time to the most recent common ancestor of currently known strains. So, author should rewrite paragraphs concerning tMRCA in a correct way.
  3. The hypothesis of the first period of TMUV evolution in mosquitos is only a hypothesis and authors should describe it like their assumption (hypothesis).
  4. It makes no sense to specify so many digits after the dot in tMRCA (138.733) if the estimated date varies between 1887 and 1982. Please, remove them from 1887.8842 and 1982.6683. It is pointless to predict this to the minute.

Illustrative material.

  • All figures with trees in the manuscript are blind. Please, add geographical and time information, not only GenBank numbers.
  • Please, add the table of strains in the supplementary (as Table 1S).
  • Try to add WNV sequences to the trees in Figure 3A-B. It can be useful.

English needs polishing or even rewriting (some sentences). Professional proof-reading is required.

Minor remarks.

Introduction, L.31-32. The TMUV genome contains a single positive-stranded RNA of approximately 11 kb…

Introduction, L.63-64. This sentence is a someway repeat of the sentence in L.61-62.

It is necessary to note that TMUV is a mosquito-borne virus in the Introduction, not Discussion.

Results, L. 241. Give a reference on the long term prevalence.

Round 2

Reviewer 1 Report

Thank you for the tailored manuscript. However, there are still many things to be addressed.

-Line 27: The sentence should be readdressed. It looks like TMUV was first isolated in China.

-BSP was not explained in the methods section.

-Line 134: “…it remained at higher level”. Should be changed. Higher than what? Higher 2009 but after 2010, constant. The sentence should be readdressed.

-Line 214-222: You did not download viruses but virus sequences. “…TMUV showed high confidence, do you mean high similarity? The whole paragraph must be readdressed.

-Line 231: What different isolates of TMUV was used? How many of them?

-Line 232-233: Which host? How are NS3 and NS5 related to host immunity? This conclusion must be readdressed.

Line 277: “Secondly, TMUV is constantly adapting to new hosts” How did you have this conclusion? What is the reference of this observation? Line 278-179 does not prove host adaptation.

Line 297-298: Not clear what the authors mean. Any reference?

Author Response

    Thank you very much for the chance you gave us to revise our manuscript. We thank the reviewers for the time and effort that they have put into reviewing the previous version of the manuscript. We admire your expertise and patience. The comments are very useful for our research and publication. We decide to accept all the comments and revise our manuscript carefully according to your comments. The changes were listed blew point by point, and were marked in red in the revised manuscript. The manuscript has greatly benefited from these invaluable suggestions.

    Based on the instructions provided in your letter, we uploaded the file of the revised manuscript. Appended to this letter is our point-by-point response to the comments raised by the reviewers.

We look forward to working with you to move this manuscript closer to publication in Viruses.

Reviewer 1

-Line 27: The sentence should be readdressed. It looks like TMUV was first isolated in China.

Clarification of Comment---TMUV was considered to associate with egg production decline and retarded growth in ducks in 2010 in China. This is the first description of TMUV related to duck diseases. The relevant description has been made on lines 28-29 of the revised manuscript.

-BSP was not explained in the methods section.

Thanks for your suggestions, the relevant descriptions have been inserted into lines 93-97 of the revised manuscript.

-Line 134: “…it remained at higher level”. Should be changed. Higher than what? Higher 2009 but after 2010, constant. The sentence should be readdressed.

Thanks for your suggestions, the modifications have been made on lines 136-139 of the revised manuscript.

-Line 214-222: You did not download viruses but virus sequences. “…TMUV showed high confidence, do you mean high similarity? The whole paragraph must be readdressed.

Thanks for your suggestions, the modifications have been made on lines 222-228 of the revised manuscript.

-Line 231: What different isolates of TMUV was used? How many of them?

Thanks for your suggestions. A total of 110 TMUV strains were used for analysis of selection pressure, the modification has been made on lines 239-240 of the revised manuscript. Furthermore, the information of the 110 TMUV strains has been inserted into Supplementary Table 1 of the revised one.

-Line 232-233: Which host? How are NS3 and NS5 related to host immunity? This conclusion must be readdressed.

Thanks for your suggestions, the modifications have been made on lines of 245-248 of the revised manuscript.

Line 277: “Secondly, TMUV is constantly adapting to new hosts” How did you have this conclusion? What is the reference of this observation? Line 278-179 does not prove host adaptation.

Thanks for your suggestions, the modification has been made on lines 289-290 of the revised manuscript.

Line 297-298: Not clear what the authors mean. Any reference?

Thanks for your suggestions. In this study, we found that cluster 2.2 is the current prevalent strain of TMUV, meanwhile other species of TMUV also exist in cluster 2.2. The phylogenetic analysis shows that cluster 2.2 has some specific amino acid mutation sites, and these mutations may be associated with the widespread and species-transmission of the cluster 2.2 strains. This phenomenon is very common in influenza virus. The relevant modification has been made on lines 306-308 and the added reference has been inserted into lines 527-529 of the revised manuscript.

Reviewer 2 Report

The authors supplemented the list of used sequences and recalculated the rate of molecular evolution and tMRCA. In addition, the text of the manuscript has been significantly improved. However, some figures and tables have disadvantages.

-          Table S1 (quite important table): Abbreviations in the right column should be deciphered (under the table).

-          Figures with phylogeny and phylogeography: The captions to the drawings should be supplemented with all the necessary information.

-          In addition, the authors should discuss the possible contribution of other genes to the evolution of the virus. It should be taken into account the possibility that the rate of evolution,  tMRCA and evolution history, can be changed if full-genome sequences would be involved in the study.

Author Response

    Thank you very much for the chance you gave us to revise our manuscript. We thank the reviewers for the time and effort that they have put into reviewing the previous version of the manuscript. We admire your expertise and patience. The comments are very useful for our research and publication. We decide to accept all the comments and revise our manuscript carefully according to your comments. The changes were listed blew point by point, and were marked in red in the revised manuscript. The manuscript has greatly benefited from these invaluable suggestions.

    Based on the instructions provided in your letter, we uploaded the file of the revised manuscript. Appended to this letter is our point-by-point response to the comments raised by the reviewers.

We look forward to working with you to move this manuscript closer to publication in Viruses.

Reviewer 2

The authors supplemented the list of used sequences and recalculated the rate of molecular evolution and tMRCA. In addition, the text of the manuscript has been significantly improved. However, some figures and tables have disadvantages.

     Table S1 (quite important table): Abbreviations in the right column should be deciphered (under the table).

     Thanks for your suggestions. These abbreviations have been deciphered under the Table S1 of the revised manuscript.

      Figures with phylogeny and phylogeography: The captions to the drawings should be supplemented with all the necessary information.

      Thanks for your suggestions. We have added to necessary information to the captions of the figures with phylogeny and phylogeography, the relevant descriptions have been inserted into lines 184-189 and lines 216-219 of the revised manuscript.

      In addition, the authors should discuss the possible contribution of other genes to the evolution of the virus. It should be taken into account the possibility that the rate of evolution, tMRCA and evolution history, can be changed if full-genome sequences would be involved in the study.

    Thanks for your suggestions. In this study, we chose the complete genome for analysis because the complete genome could better reflect the evolutionary process of the TMUV population, while a single gene could not better reflect the evolution of the entire population due to some limitations. As suggested by the reviewer, the discussion of full-genome sequences and other genes has been inserted into lines 283-285 of the revised manuscript.